# BSC-Net: Background Suppression Algorithm for Stray Lights in Star Images

**Yabo Li, Zhaodong Niu \*, Quan Sun, Huaitie Xiao and Hui Li**

National Key Laboratory of Science and Technology on ATR, College of Electronic Science and Technology, National University of Defense Technology, Changsha 410073, China
\* Correspondence: niuzd@nudt.edu.cn

**Abstract:** Most background suppression algorithms are weakly robust due to the complexity and fluctuation of the star image's background. In this paper, a background suppression algorithm for stray lights in star images is proposed, which is named BSC-Net (Background Suppression Convolutional Network) and consist of two parts: "Background Suppression Part" and "Foreground Retention Part". The former part achieves background suppression by extracting features from various receptive fields, while the latter part achieves foreground retention by merging multi-scale features. Through this two-part design, BSC-Net can compensate for blurring and distortion of the foreground caused by background suppression, which is not achievable in other methods. At the same time, a blended loss function of smooth_L1&Structure Similarity Index Measure (SSIM) is introduced to hasten the network convergence and avoid image distortion. Based on the BSC-Net and the loss function, a dataset consisting of real images will be used for training and testing. Finally, experiments show that BSC-Net achieves the best results and the largest Signal-to-Noise Ratio (SNR) improvement in different backgrounds, which is fast, practical and efficient, and can tackle the shortcomings of existing methods.

**Keywords:** star image; deep learning; background suppression; stray light





## 1. Introduction

With the expansion of human space activities, the quantity of space debris is growing at the same time [1,2]. In order to surveil space debris, the star images are collected by CCD devices. However, the backgrounds in these pictures have an impact on the surveillance of space debris. To address this issue, numerous algorithms for the background suppression of star images have been proposed.

### 1.1. Literature Review

These algorithms can be roughly divided into two categories: traditional methods based on features of star images and deep learning methods based on neural networks.

The traditional methods were proposed earlier, and the star images were initially processed by conventional approaches such as mean filtering, wavelet transform, contour extraction and 3D matched filtering [3–6]. There is also some astronomy software available, such as SExtractor [7]. However, because these methods are not universal; there are problems such as blurred details and significant mistakes in star images, and several targeted algorithms have then been brought up. Wang et al. (2012) proposed a suppression algorithm for background clutter in star images. It is based on neighborhood filtering and can retain the characteristics of weak targets and star edges [8]. Yan et al. (2014) devised a stray light stripe noise correction algorithm on the basis of stray light spatial distribution characters on CCD/TDI-CCD focal plane. The generalized noise value of stray light stripe noise removed images is below 5%, which is satisfied with the need of remote sensing imagery radiometric correction precision [9]. Chen et al. (2015) proposed a method that

adopts multi-structural elements in morphological filters to suppress the background and clutter in star images. The filtered image was enhanced by the plateau's histogram image equalization algorithm. After these processes, the contrast ratio of the image is improved remarkably [10]. Wang et al. (2018) proposed a method to deal with stripe nonuniformity noise and defective pixels. It used one-dimensional mean filtering, one-dimensional feature point descriptor and moment matching to effectively compensating the defective pixels [11]. Zhang et al. (2021) used the intensity prior to star maps to estimate the depth information of the atmosphere and remove the unevenly distributed stray light noise, which achieved a higher background suppression factor [12]. Zou et al. (2021) proposed a segmentation method of star images with complex backgrounds based on a correlation between space objects and one-dimensional (1D) Gaussian morphology. On the premise of using the shape of the object to achieve segmentation, the influence of the complex background is weakened [13]. Wang et al. (2022) proposed a star background parameter estimation method, which continuously discards the high-gray areas and reduces the sample variance, and the background parameters are used to suppress noise. Experiments showed that the method can filter out most of the noise and retain and enhance the target to the maximum extent under the influence of complex background noise and stars [14].

Although the traditional approaches have a solid theoretical foundation and long development history, the algorithms need to be changed in different star images processing, which leads to the poor robustness of traditional methods. Additionally, some traditional methods are very complex and computationally intensive, making them unsuitable for practical applications.

With the rapid development of deep learning, methods based on a convolutional neural network (CNN) have been introduced into the processing of star images [15–26]. Liu et al. (2019) proposed a deep learning method with the application of a residual network and down-sampling algorithm; it achieved significant improvement in the SNR of the star points [27]. Xue et al. (2020) proposed the deep learning method and pixel-level segmentation to achieve target detection of the star images [28]. Xie et al. (2020) effectively suppressed mixed Poisson–Gaussian noise in star images by taking a mixed Poisson–Gaussian likelihood function as the reward function of a reinforcement learning algorithm [29]. Zhang et al. (2021) proposed a two-stage algorithm, which used the reinforcement learning idea to realize the iterative denoising operation in the star point region [30].

### 1.2. Imaging Features of Star Images

After understanding the methods of background suppression in star images, it is necessary to be familiar with the characteristics of star images. The imaging features of star images can be expressed as Formula (1) [31]:

$$F(i,j) = T(i,j) + S(i,j) + B(i,j) + N(i,j) \tag{1}$$

where $F(i, j)$ represents the pixel value at position $(i, j)$ in the star image, $T(i, j)$ is the space debris or space targets, $S(i, j)$ is the star, $B(i, j)$ is the undulating background in the image and $N(i, j)$ is the noise. Among the four components, space debris and stars are collectively called foreground, while modulated background and noise are collectively called background.

In the imaging process of star images, the modulated background has the greatest influence on image quality, which is correlated with the imaging part. In space-based images, sunlight, moonlight and Earth stray light are the main sources of stray light. The moonlight is formed by the reflection of sunlight by the moon. The Earth stray light is a kind of radiant light that impacts into the atmosphere and is returned to the cosmic space by the scattering of the atmosphere and the reflection of clouds and ground. In ground-based images, the interference from sunlight and moonlight still exists, but the Earth's stray light consists of scattered light and diffuse light from the Earth's surface and atmosphere. High-speed moving objects and clouds are also classified as Earth's stray light.

Sunlight is a strong interference light source, whereas moonlight is mainly manifested as a highlighted area that gradually spreads to the surroundings with a certain point as the center. The atmospheric light will cause local highlights, interference fringes and partial occlusions in the image [32–34].

Figure 1 shows the star images with different backgrounds and their 3D grayscale images. In Figure 1a, there is a highlighted area at the top edge. In Figure 1b, there is locally highlighted stray light interference at the upper left corner, which comes from the moonlight. As can be seen from Figure 1(b-1), the existence of stray light makes the gray level of the image fluctuate greatly, and the contrast of foreground and background is low. In Figure 1c, there is cloud cover. The linear fringe interference in Figure 1d comes from object that is moving at high speed in the sky. The background distributions of Figure 1e,f are relatively uniform, and the background's gray value is low.

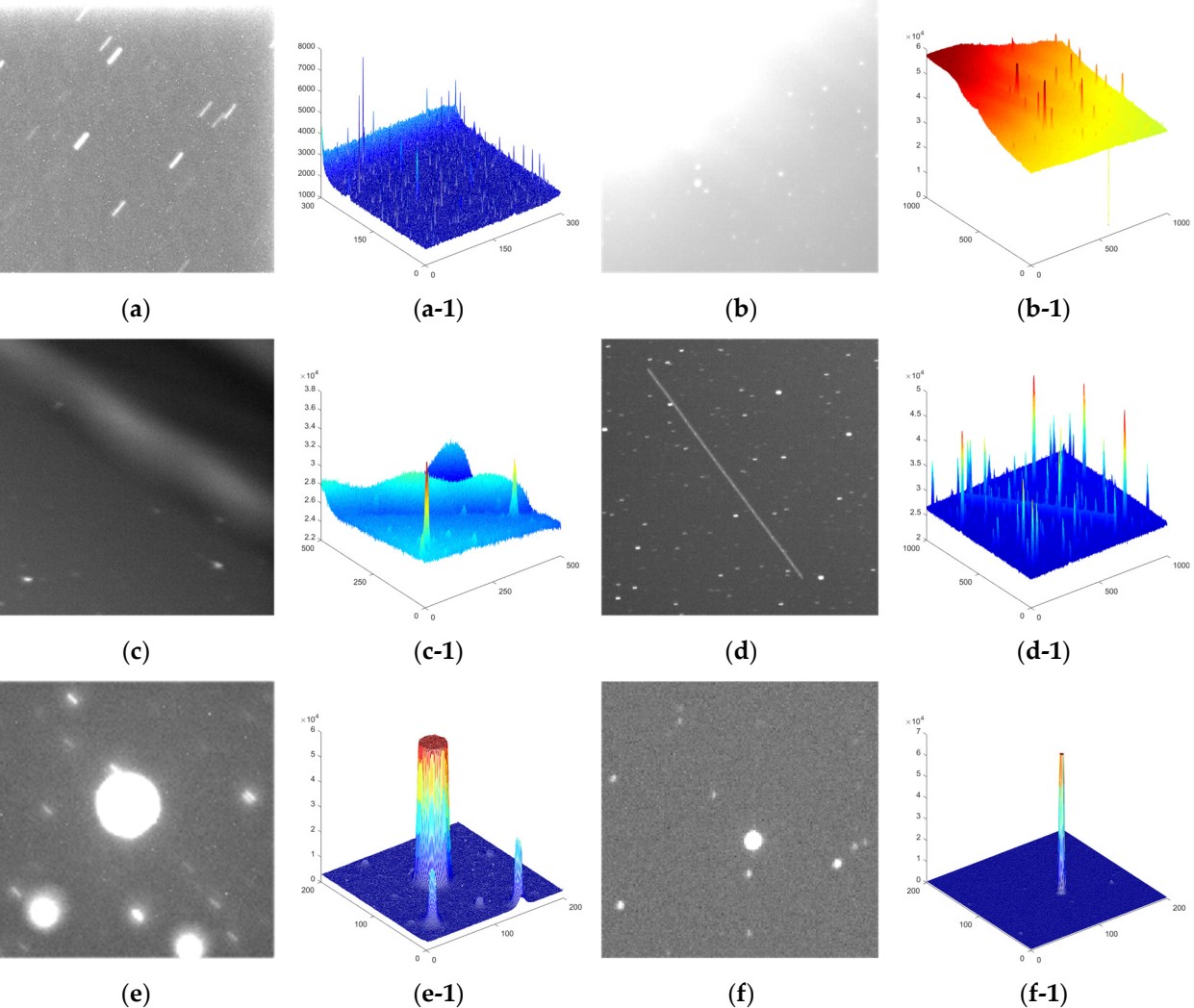

**Figure 1.** Schematic diagram of various stray lights. (**a**) Highlight stray light; (**a-1**) 3D grayscale image of (**a**). (**b**) Moonlight stray light. (**b-1**) 3D grayscale image of (**b**). (**c**) Earth stray light. (**c-1**) 3D grayscale image of (**c**). (**d**) High-speed moving objects in the field of view (**d-1**) 3D grayscale image of (**d**). (**e**) Uniform background. (**e-1**) 3D grayscale image of (**e**). (**f**) Uniform background. (**f-1**) 3D grayscale image of (**f**).

### 1.3. SSIM and Peak Signal-to-Noise Ratio (PSNR)

Since a background suppression performance requires some metrics to evaluate images, some general metrics and concepts will be explained first.

SSIM is based on the assumption that the human visual system is highly adapted for extracting structural information from the scene, and therefore, the measure of structural similarity can provide a good approximation of the perceived image quality. SSIM takes into account the brightness, contrast and intensity scale of the two images at the same time. Its calculation formula is as follows [35]:

$$SSIM(X, Y) = \frac{(2\mu_X\mu_Y + C_1)(2\sigma_{XY} + C_2)}{(\mu_X^2 + \mu_Y^2 + C_1)(\sigma_X^2 + \sigma_Y^2 + C_2)} \tag{2}$$

where $C_1$ and $C_2$ are constants, $X$ and $Y$ represent the two images being compared, $\mu_X, \mu_Y$ are the means of the images, $\sigma_X^2, \sigma_Y^2$ are the variances of the images and $\sigma_{XY}$ is the covariance. Figure 2 shows images with different SSIM values. As can be observed, when the brightness, contrast and general distribution of the image are more similar to the reference image, the SSIM value is higher. Through numerous experiments and observations, it is determined that, when the SSIM value exceeds 0.850, the image is quite near to the reference image.

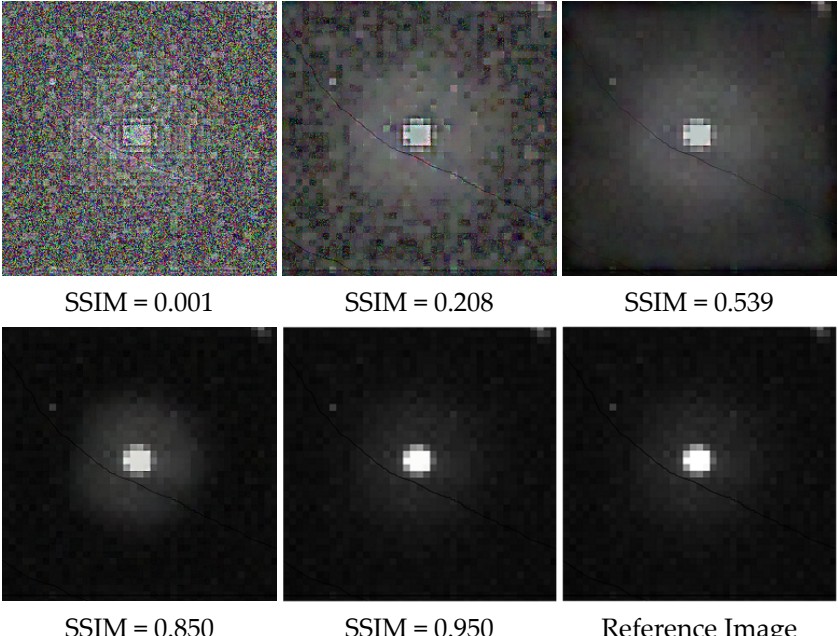

SSIM = 0.001  SSIM = 0.208  SSIM = 0.539

SSIM = 0.850  SSIM = 0.950  Reference Image

**Figure 2.** Images of different SSIM values.

PSNR is an important indicator for evaluating image quality, and its calculation formulas are as follows:

$$MSE = \frac{1}{mn}\sum_{i=0}^{m-1}\sum_{j=0}^{n-1}[I(i,j) - K(i,j)]^2 \tag{3}$$

$$PSNR = 10 \cdot \log_{10}(\frac{MAX_I^2}{MSE}) = 20 \cdot \log_{10}(\frac{MAX_I}{MSE}) \tag{4}$$

where $I$ represent the evaluated output results, and $K$ represents the reference image. $MAX_I$ is the maximum value of the image pixel. In this paper, the maximum value of the 16-bit integer unsigned image is 65,535. According to Formula (3), when the two images have high similarity, the $MSE$ is smaller, and the PSNR increases accordingly. Therefore, the larger the PSNR, the closer the image is to the reference image.

Through understanding the existing methods and basic concepts, we found that, even though the deep learning approach has not been widely accepted in the background suppression of star images, it is clear from the literature and experiments that these approaches perform well on our issue. Since most of the existing literature uses simulated data, there is

the problem of a single background type, which makes the trained network unsuitable for images with complex backgrounds. Therefore, a deep learning algorithm that can handle real images well is needed.

In this paper, a background suppression algorithm primarily implemented for suppressing stray lights in star images is proposed, named BSC-Net. This algorithm adopts an end-to-end network structure without the need for image preprocessing and manual feature construction. Network training does not use simulation data but real data. Benefitting from the introduction of the blended loss function of smooth_L1&SSIM, the image after background suppression does not show evident distortion. We tested the algorithm on the test sets composed of real images to obtain quantitative results.

The rest of the article includes the following contents. In the second section, the network structure, key algorithm of BSC-Net and dataset preparation process are described. In the third section, both the comparative experimental results and the quantitative evaluation tables are presented. In the fourth section, the research results are analyzed, and the reasons are given according to the algorithm principle. In the last section, the conclusion will be given.

## 2. Methods

In this section, a background suppression algorithm mainly for stray light in star images is proposed, which is named BSC-Net and consists of two parts in one network. At the same time, the blended loss function of smooth_L1&SSIM will be suggested in this section, which can both hasten the network convergence and avoid image distortion. Based on BSC-Net and the blended loss function, the dataset consisting of real images will be used for training and testing. After the training process, an end-to-end model will be produced to deal with the noisy star images.

### 2.1. Network Structure of BSC-Net

The network structure of BSC-Net is shown in Figure 3. The structure is divided into two parts [36]. The blue part on the left is the background suppression part, which is responsible for extracting the features under different receptive fields to suppress the background. The orange part on the right is called the foreground retention part, which preserves the foreground by merging the features under different receptive fields. The blue and orange squares represent the feature map of each step, and the reported value represents the number of feature map channels; the minimum is 1, and the maximum is 512.

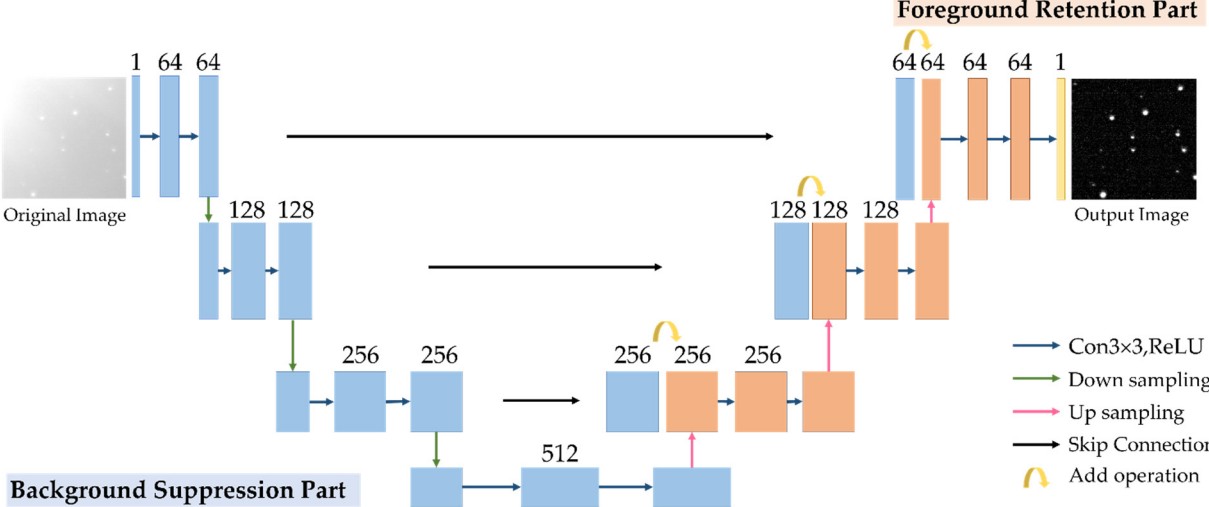

**Figure 3.** The network structure diagram of BSC-Net.

In order to better understand the composition and function of the network, each part will be described separately.

### 2.1.1. Background Suppression Part

Down-sampling and convolution comprise the majority of the background suppression part. Down-sampling can expand the receptive field and obtain information in various scales. Figure 4 depicts how the feature map changes during multiple down-sampling operations. It can be seen that, when the receptive field widens, the red area expands to the surrounding area, and the details in the image gradually decrease.

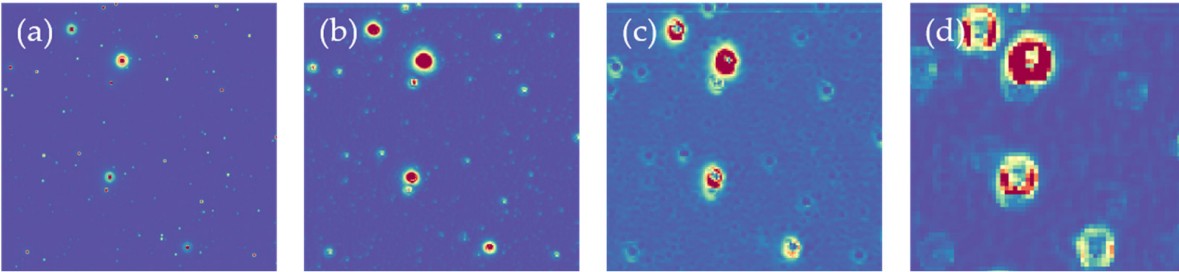

**Figure 4.** Changes in feature maps during down-sampling. (**a**) The feature map of the input image after convolution. (**b**) The feature map after the first down-sampling. (**c**) The feature map after the second down-sampling. (**d**) The feature map after the third down-sampling.

Convolution is a kind of weighted summation process. A basic Gaussian convolution kernel can smooth out noise in an image, while numerous convolution kernels can form a wide range of filters to handle various foreground and background types. Figure 5 shows the images before and after convolution. It is found that convolution has a certain effect on smoothing the stray light in the background of a star image.

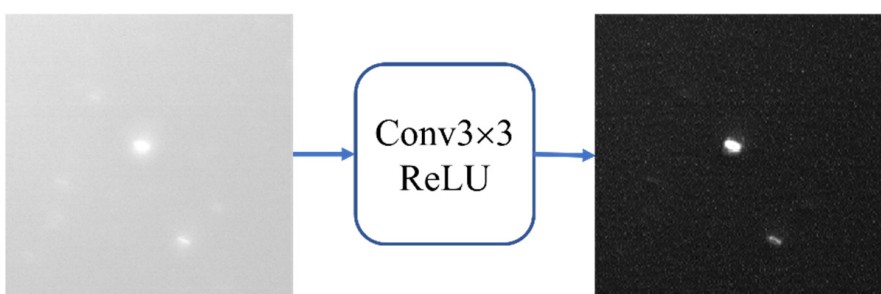

**Figure 5.** Changes in the image before and after the convolution operation.

### 2.1.2. Foreground Retention Part

Up-sampling, skip connection and convolution comprise the majority of the foreground part. Among them, up-sampling and skip connection play an important role in the foreground preservation of a star image. A portion of Figure 3 is enlarged in Figure 6, which shows the specific process in this step. Figure 4 has already explained that down-sampling will cause the loss of details, so the details need to be supplemented next. First, up-sampling doubles the size of feature maps while it halves the number of channels. Then, the up-sampled feature map and the same-scale feature map are numerically added to the corresponding channel to create a new feature map, which is then convolved again. The expected outcome of this operation is presented in Figure 7.

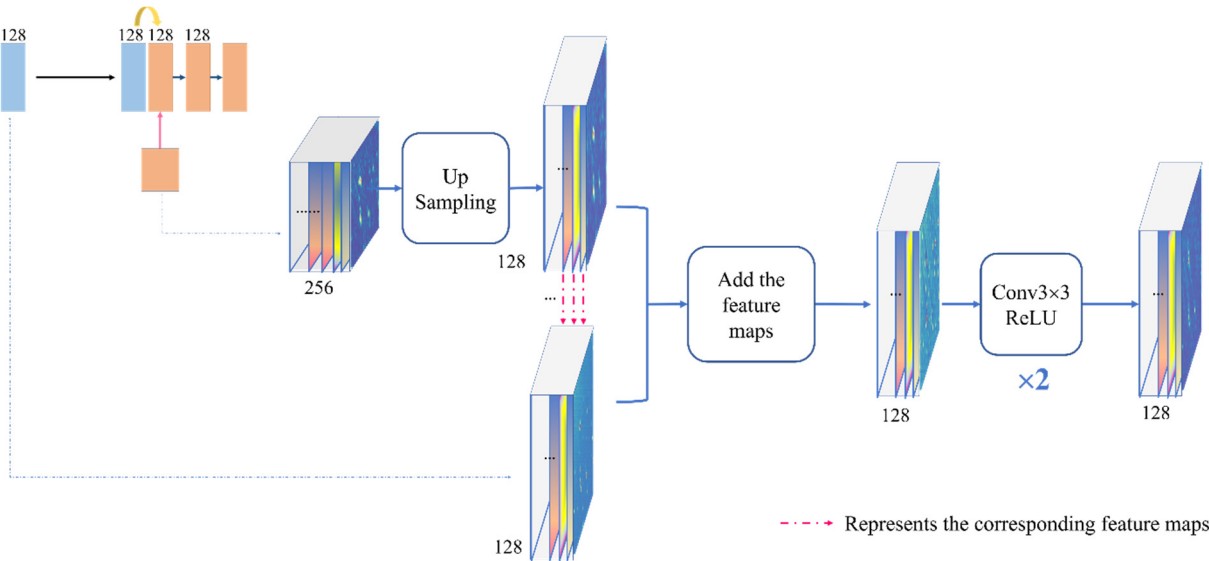

**Figure 6.** Schematic diagram of the up-sampling and skip connection process.

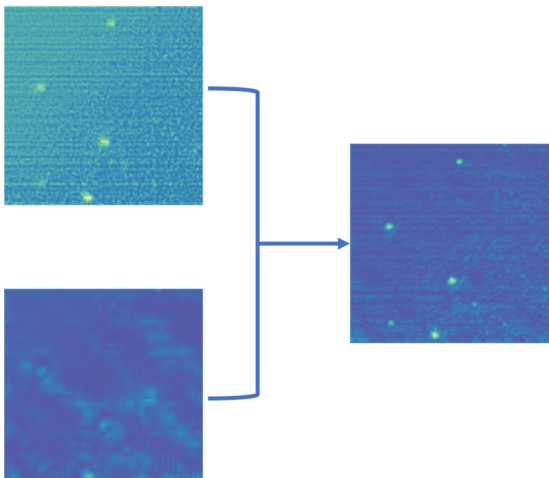

**Figure 7.** Expected effect of a skip connection.

### 2.1.3. Strengths of BSC-Net

The main structure and details of BSC-Net are described above. Through these designs, several advantages of the algorithm are summarized as follows:

- The selected structure cleverly combines the functions of background weakening and foreground preservation of two parts in one network, which can compensate for blurring and distortion of the foreground caused by the background suppression part. Unlike our approach, existing algorithms often only concentrate on one of the two functions.

- Compared with the majority of convolutional networks, BSC-Net significantly reduces the number of convolutional layers. Except for the final output layer, each of the other convolutional layers only has two layers. On the one hand, too many convolution operations will increase the amount of computations, thus affecting the processing efficiency of the network. On the other hand, the receptive field reached 68 due to the down-sampling and convolution processing in the background suppression part, which is sufficient to handle the stars and targets in the range of $3 \times 3$–$20 \times 20$.

- BSC-Net does not require image preparation, size restriction and manual feature construction. After BSC-Net processing, a clean image with background suppression is output with the size and dimensions unchanged.

*2.2. Blended Loss Function of Smooth_L1&SSIM*

Based on the network structure of BSC-Net and the features of star images, the corresponding loss function is designed. The loss function is used to measure the error between the predicted and true values. Since the star image is a 16-bit grayscale image, the commonly used loss functions do not completely match the image. In order to make the results obtained by the network closer to the optimal solution, this paper proposes a blended loss function combining SSIM and smooth_L1 [37]:

$$Loss = (1 - A) \times smooth_{L1}(y - \hat{y}) + A \times [1 - SSIM(y, \hat{y})] \tag{5}$$

The blended loss function consists of two loss functions, smooth_L1 and SSIM. $y$ represents the ground truth, and $\hat{y}$ represents the predicted value from the network. $A$ is used as an adjustment parameter to adjust the proportion of SSIM in the loss function calculation process.

The calculation formula of smooth_L1 is as follows [37]:

$$smooth_{L1}(x) = \begin{cases} 0.5x^2 & if |x| < 1 \\ |x| - 0.5 & otherwise \end{cases} \tag{6}$$

$$\text{L1\_LOSS} = \frac{1}{n}\sum_{i=1}^{n} |x| \tag{7}$$

$$\text{L2\_LOSS} = \frac{1}{n}\sum_{i=1}^{n} x^2 \tag{8}$$

$x$ represents the error between the predicted value and the ground truth. Smooth_L1 combines the benefits of L1_LOSS and L2_LOSS, whose definitions can be seen in Formulas (7) and (8). Smooth_L1 can prevent a gradient explosion caused by outliers and has a stable solution procedure, making it more appropriate for star images. Since our training samples come from different datasets, the mean value of each image fluctuates between 2000 and 30,000, and there are also regions in the image where the gray value changes drastically. The features of smooth_L1 make it suitable for our images.

SSIM is structurally similar; the calculate formula is the same as Formula (2). In the star image, the way that human eyes distinguish stars, space debris and backgrounds is by brightness, contrast and grayscale. However, smooth_L1 only judges the differences in pixel values, which may lead to distortions of the edges. Therefore, we choose SSIM as part of the loss function to prevent these phenomena [38,39].

The weighting method of the loss function follows the changing trend of smooth_L1 and SSIM. When the image is closer to the ground truth, the value of smooth_L1 becomes smaller, and the value of SSIM becomes larger. After weighting, the predicted image is closer to the true value, and the overall loss becomes smaller.

*2.3. Dataset Preparation of Real Images*

Since this paper uses real images as training data and test data, it is necessary to choose suitable real images and obtain clean images as ground truths through image processing.

Most of the data is collected by Dhyana 4040 with fixed settings (see more details in Appendix A). Part of the data is separated as a test set, while the other part is divided into three categories and added to the training set. The three categories are: (1) the background fluctuation is small, and the grayscale is small at the same time. (2) The background fluctuates greatly, and there is local stray light. (3) The background fluctuation is small, but the background's overall grayscale is large.

Based on state-of-the-art methods, we select morphological transform [27] as the way to prepare the dataset. At the same time, this approach will be used in the next comparison experiments. Clean images produced by this method will serve as reference images in image evaluation. Figure 8 shows the process of getting a clean image from a real image.



Noisy Original Image　　　　　　　　　　　　　　　　　　Clean Output Image

**Figure 8.** The process of getting a clean image from a real image.

## 3. Results

In this section, we will demonstrate the background suppression performance of BSC-Net through a series of experimental results. First, the intermediate results of the network will be visualized to verify the feasibility of the algorithm. Subsequently, BSC-Net is compared with existing methods, and the SNR of dim targets is calculated to prove the superiority of the algorithm. Finally, the quantitative evaluation data will be presented.

### 3.1. Experimental Environment

The experiment uses the Ubuntu18.04 operating system and a Pytorch deep learning development framework. We use Python as the development language. The CPU used in the experiment is an Intel Core i9-9940X@3.3GHz, and the GPU is NVIDIA GeForce RTX 2080Ti. Adam is used as the optimizer in the training process, the batch size is set to 8 and the crop size for each sample is $128 \times 128$ pixels. The initial learning rate is set to 0.0003, cosine annealing is used as the learning rate optimization method, the minimum learning rate is set to 0.00003 and the epoch is 2000.

### 3.2. Evaluation Criteria of Background Supprssion Effectiveness

Section 1.2 describes the characteristics of star images and defines the properties of foreground and background. Section 1.3 explains some general metrics and concepts. Then, an evaluation metric for the background suppression algorithm is given. Note that the performance of the background suppression is measured also by the corresponding enhancement of the foreground. We propose three different assessment indicators: SNR of targets, SSIM of images and PSNR of images.

#### 3.2.1. SNR of Targets

The first index is the SNR of the targets. The increase of SNR indicates the contrast between the foreground and the background is higher, which benefits from the adequate suppression of the background.

In star images, the SNR improvement of single target dominates our attention. As a result, the conventional SNR calculation approach will not be employed. Since the test images are real images, the ranges of the foreground and background need to be determined first. The calculation method of SNR is divided into the following five steps:

1.　First, the evaluation range of single target is determined, and the pixels in that range are called $I$. A threshold segmentation of $I$ will be done.

$$\begin{cases} I_f = 1 & I \geq threshold \\ I_f = 0 & I < threshold \end{cases} \tag{9}$$

where $I_f$ is the foreground mask obtained by threshold segmentation, and the *threshold* is typically determined by the mean and variance.

2.　$I_f$ is dilated to remove the effect of the transition region, which contains uncleared stray light around the target.

$$I_f \oplus B = \{ z | (\hat{B})_z \cap I_f \neq \varnothing \} \tag{10}$$

$B$ is set to be a structure element of size 5. The area after the dilation operation is called $I_f^{'}$.

3. The mask of the background is determined by inverting $I_f^{'}$, which is called $I_b$.

$$\begin{cases} I_b = 1 & I_f{'} = 0 \\ I_b = 0 & I_f{'} = 1 \end{cases} \tag{11}$$

4. $I_f$ and $I_b$ are matched with $I$, respectively, to obtain the foreground region $I_B$ and background region $I_F$.

$$\begin{cases} I_b \& I = I_B \\ I_f \& I = I_F \end{cases} \tag{12}$$

5. Calculating SNR.

$$SNR = \frac{\mu_{I_F} - \mu_{I_B}}{\sigma_{I_B}} \tag{13}$$

where $\mu_{I_F}$, $\mu_{I_B}$ represent the mean of the foreground and background, respectively, and $\sigma_{I_B}$ is the variance of the background.

### 3.2.2. SSIM and PSNR

In the latter two categories of the evaluation metric, we regard that both the experimental results and the reference image as the estimation of the foreground and assess them by SSIM and PSNR values. The formulas refer to Section 1.3. High SSIM and PSNR equate to a high similarity, which is obtained by accurate foreground estimations and sufficient background suppression. Therefore, there is a positive correlation between the SSIM&PSNR value and background suppression effectiveness, making them become credible evaluation indicators.

### 3.3. Verification Experiment of Network Function

In the process of network design, we divided the network into a background suppression part and foreground preservation part. In order to intuitively show the process of the network and reflect the rationality of the BSC-Net design, the intermediate results of the network are output.

Figure 9 reflects the feature maps at different stages of the network, which are obtained by averaging multiple channels at processing stages. Among them, Figure 9a–d are the feature maps of the background suppression part, and Figure 9e–g are the feature maps of the foreground retention part. Through the change trend of the feature maps, it can be confirmed that:

1. The background suppression part of BSC-Net can achieve the expected effect. From Figure 9a–d, through down-sampling and convolution, the receptive field increases, the background in the image is gradually uniformized and the brightness is gradually weakened. Detail information in the foreground is reduced.
2. The foreground retention part of BSC-Net can achieve the expected effect. From Figure 9e–g, through up-sampling and the skip connection, the detail information is completed, the foreground is preserved and the background is further suppressed by convolution. At the end, we get a clean output image.

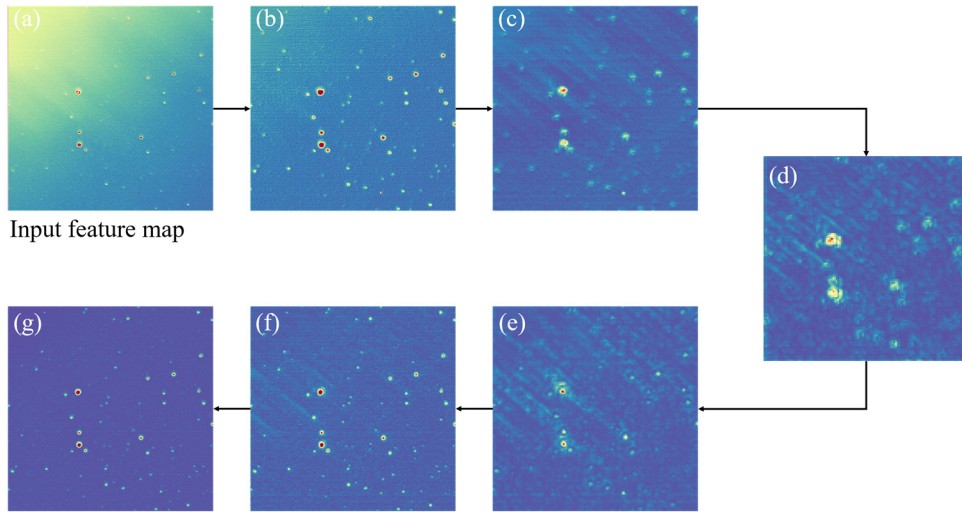

Input feature map

Output feature map

**Figure 9.** Intermediate results of the BSC-Net processing images. (**a**) Input feature map. (**b**) Feature map with 128 channels. (**c**) Feature map with 256 channels. (**d**) Feature map with 512 channels. (**e**) Feature map with 256 channels after skip connection. (**f**) Feature map with 128 channels after skip connection. (**g**) Output feature map.

*3.4. Contrastive Experiment of Suppressing Stray Light*

To test the performance of the proposed algorithm, different types of images in the test sets are tested separately. We will compare our algorithm with other six methods: median filtering [3], BM3D [5], Top-Hat transform [40], morphological transform [41], DCNN [27] and SExtractor [7]. The first three of them are conventional background suppression or denoising algorithms, DCNN is a deep learning method and morphological transformation is the dataset preparation method. Source Extractor (SExtractor) is a commonly used astronomy software. For all experimental results, the Root Mean Square (RMS) of the background is given for the reference (see Appendix A for more details).

Since this paper mainly focuses on stray light's suppression and SNR improvement of the star point, several typical stray light images are selected for display; they are: local highlight stray light, linear fringe interference and clouds occlusion stray light.

1.　Local highlight stray light: Figure 10 shows the results of local highlight stray light in different algorithms and the zoomed-in view of the dim target at the same location. It can be seen that BM3D and the median filter do not entirely eliminate background stray light. In BM3D, the average gray value of backgrounds is higher than that of the target, resulting in a negative SNR value. Top-Hat and DCNN over-eliminate the target and background, so the target information is almost lost, bringing the SNR value close to zero. The morphological approach and SExtractor both show positive results, but in terms of target retention, BSC-Net outperforms them.

2.　Linear fringe interference: Figure 11 shows the results of linear fringe interference and the zoomed-in views. It can be found that the median filter and BM3D treat the target and the stray light as a whole while their gray values are similar, so that the contrast is reduced, and the SNR is lower than the original image. The target's form is altered by Top-Hat and DCNN, while the high background variance causes the SNR to decrease. SExtractor suppresses most of the background, except for the fringe interference, but its effect is not significant in improving the target SNR. The stray light background is effectively suppressed by both the morphological method and BSC-Net, but BSC-Net has more advantages in terms of the SNR value.

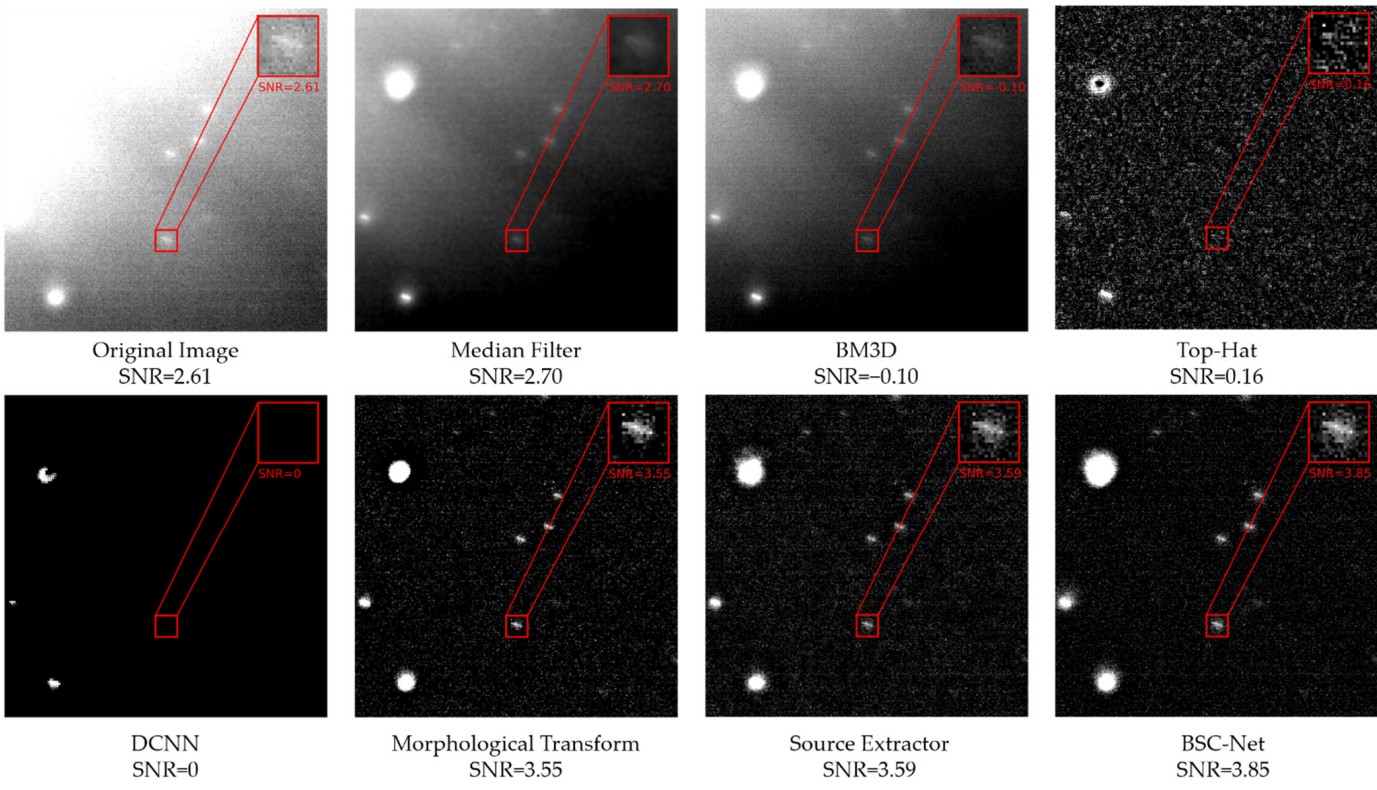

**Figure 10.** Results of the local highlight stray light in different algorithms (RMS = 226.55).

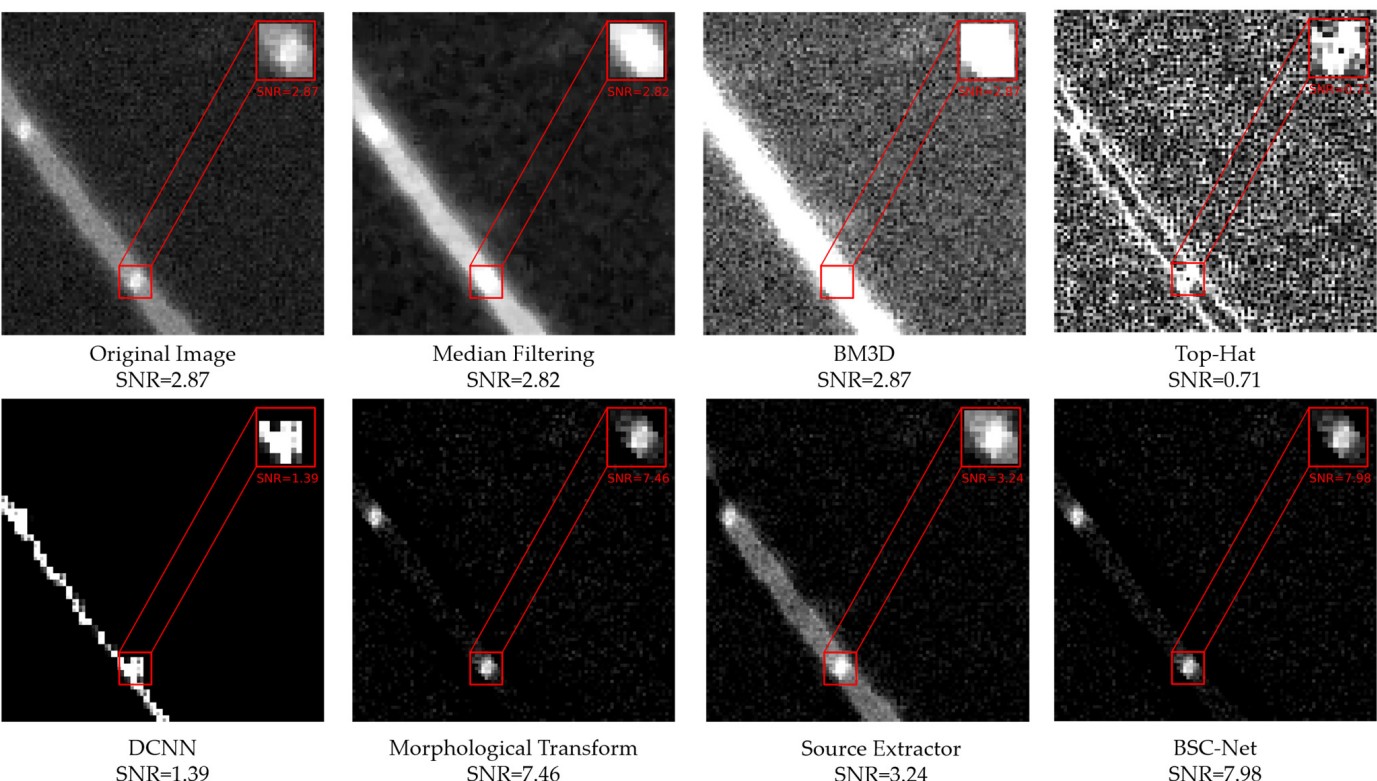

**Figure 11.** Results of linear fringe interference in different algorithms (RMS = 269.41).

3.  Clouds occlusion stray light: Figure 12 shows the results of clouds covering stray light and the zoomed-in views. In the figure, the median filter and BM3D both improve the SNR and make the distinction between foreground and background more evident, but the suppression effect of the two on cloud occlusion is weak. The Top-Hat algorithm is less robust for this kind of image, as the target information is lost, and the SNR is near to zero. While DCNN reduces the stray light, it also eliminates foreground clutter. The morphological method, SExtractor and BSC-Net all suppress the stray light to a certain extent, but BSC-Net achieves better target preservation and higher SNR improvement.

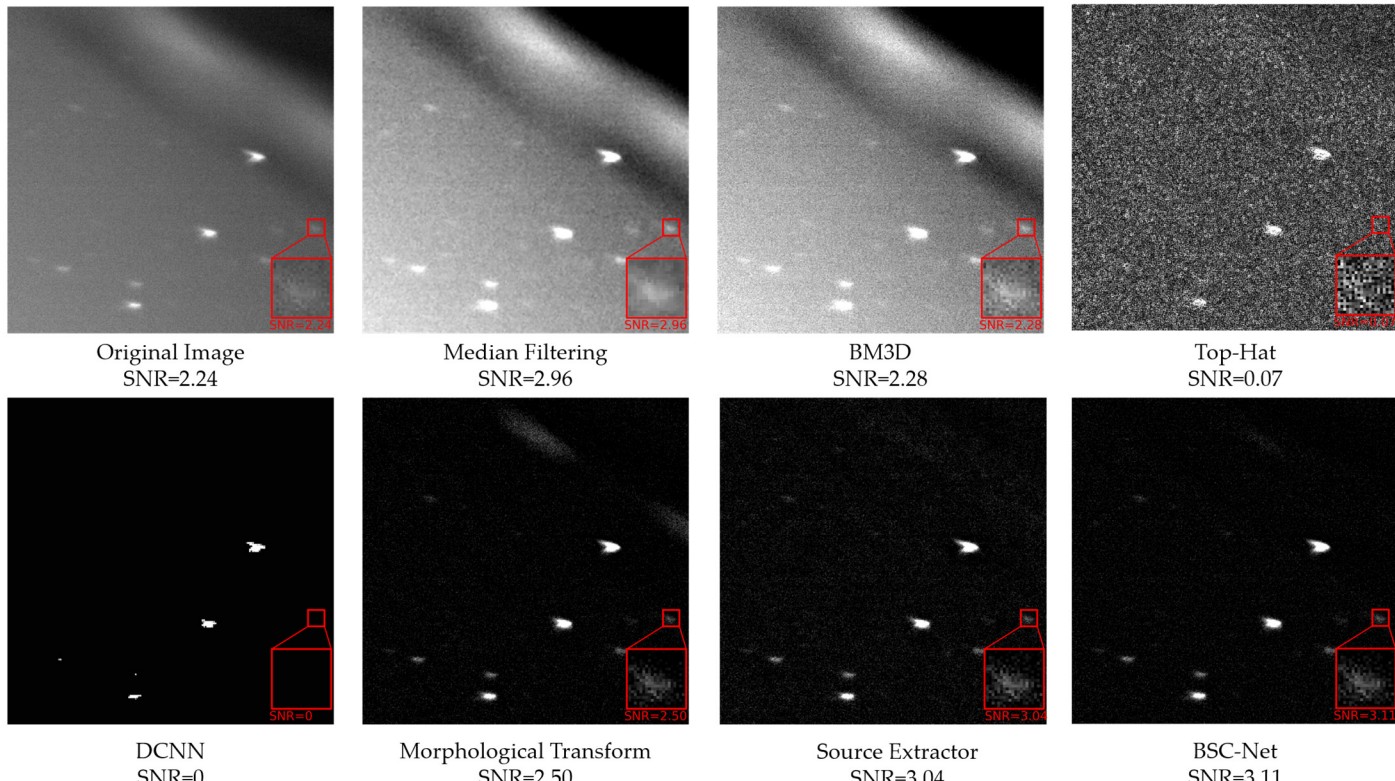

**Figure 12.** Results of clouds occlusion stray light in different algorithms (RMS = 205.97).

To examine the effects of background suppression more easily, Figure 13 depicts the matching grayscale images of Figure 12. Among the algorithms, median filtering, morphological transform, SExtractor and BSC-Net all achieve significant SNR improvement. Nonetheless, it is evident through the grayscale images that a median filter only smooths out the undulating backdrop and makes the variance value smaller; it does not achieve an evident change in the overall grayscale. On the contrary, the other three methods both greatly decrease the background's gray level, making the foreground more noticeable. When it comes to the complex cloud occlusion part, BSC-Net obviously accomplishes a better result by reducing the background fluctuation greatly.

Based on the above comparative experiments, BSC-Net achieves the best results in three typical stray light images, which can not only suppress the undulating background but also preserve the foreground details. Moreover, BSC-Net has a strong robustness and can deal with real images with complex backgrounds.

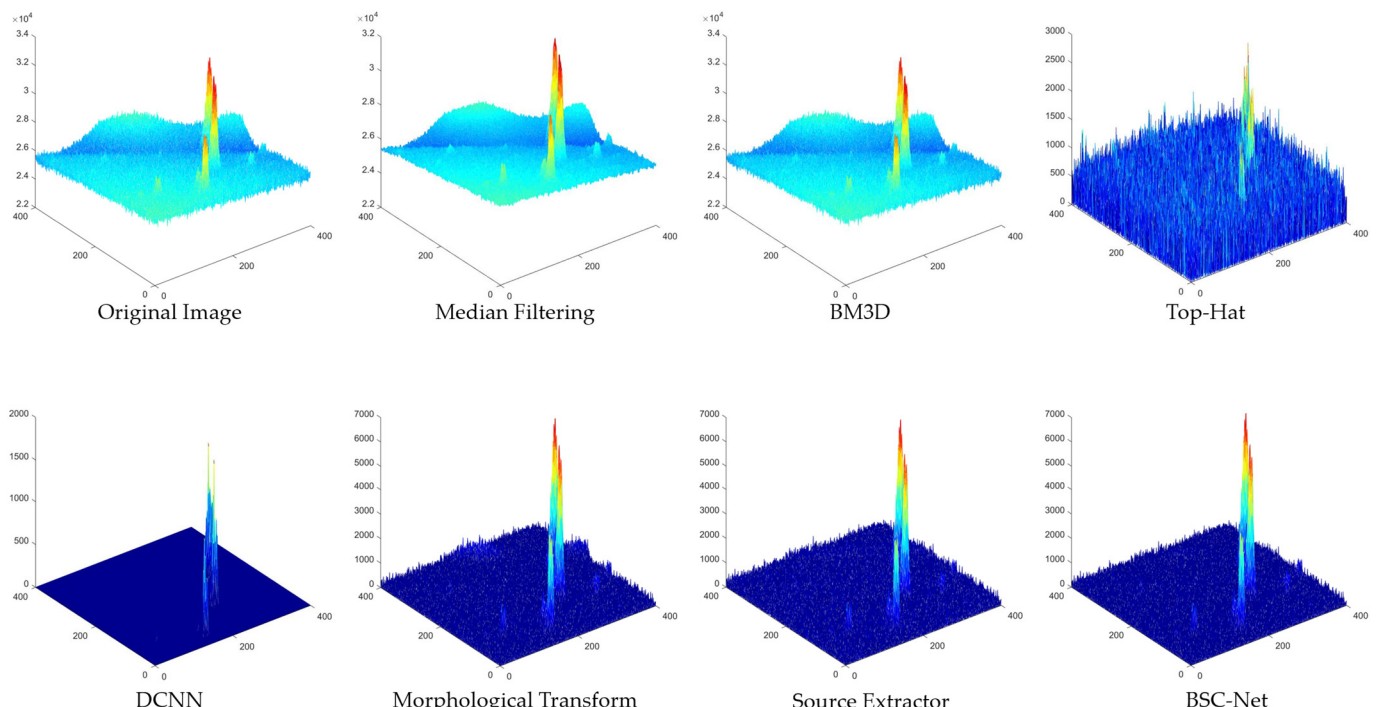

**Figure 13.** Grayscale image of cloud occlusion stray light in different algorithms.

### 3.5. Quantitative Evaluation Results for Different Datasets

To further test the performance of BSC-Net, we conducted comparative experiments on six different types of testing images and calculated the PSNR and SSIM values of the images for a quantitative evaluation. Since the reference image is required, the morphological transform method will serve as the source of the reference image.

Tables 1 and 2 show the results of a quantitative evaluation of the images. In terms of the PSNR and SSIM values of the images, the results of BSC-Net are significantly better than those of the other comparison algorithms, with an average PSNR improvement of 42.61 dB and an average SSIM of 0.8713. It can be considered that the overall image quality is high. Although the median filter and BM3D can improve the SNR of a single target, they do not significantly improve the holistic PSNR value and SSIM value. Due to the effective background suppression, Top-Hat, DCNN and SExtractor substantially increase the images' PSNR in comparison to the original images. However, because all of them destroy the foreground information, their SSIM values are low and the image quality is poor.

**Table 1.** The quantitative results on different datasets based on PSNR(/dB).

| Dataset \ Methods | Original | Median Filter | BM3D | Top-Hat | DCNN | Source Extractor | BSC-Net |
|---|---|---|---|---|---|---|---|
| Data1 [1] | 4.40 | 4.41 | 4.40 | 41.36 | 42.44 | 48.57 | **50.86** |
| Data2 [1] | 7.74 | 7.74 | 7.74 | 44.36 | 52.20 | 43.89 | **65.89** |
| Data3 [1] | 30.18 | 30.20 | 30.18 | 57.86 | 60.74 | 63.28 | **71.77** |
| Data4 [1] | 3.95 | 3.85 | 3.85 | 42.44 | 50.95 | 43.95 | **55.90** |
| Data5 [1] | 8.19 | 8.19 | 8.19 | 45.91 | 50.16 | 52.07 | **62.73** |
| Data6 [1] | 52.69 | 50.01 | 52.71 | 71.54 | 52.20 | 78.62 | **79.69** |

[1] The calculation way of PSNR was described in Section 2. The RMS of these datasets: Data1 (311.46); Data2 (269.41); Data3 (67.45); Data4 (226.55); Data5 (205.97); Data6 (9.162).

**Table 2.** The quantitative results on different datasets based on SSIM.

| Methods Dataset | Original | Median Filter | BM3D | Top-Hat | DCNN | Source Extractor | BSC-Net |
|---|---|---|---|---|---|---|---|
| Data1 [1] | 0.0045 | 0.0034 | 0.0045 | 0.1420 | 0.0015 | 0.0490 | **0.7697** |
| Data2 [1] | 0.0024 | 0.0012 | 0.0025 | 0.1402 | 0.0108 | 0.0331 | **0.8830** |
| Data3 [1] | 0.0069 | 0.0008 | 0.0045 | 0.1783 | 0.0539 | 0.0958 | **0.8905** |
| Data4 [1] | 0.0013 | 0.0005 | 0.0017 | 0.1235 | 0.0011 | 0.0396 | **0.8874** |
| Data5 [1] | 0.0034 | 0.0013 | 0.0033 | 0.1734 | 0.0003 | 0.0343 | **0.9175** |
| Data6 [1] | 0.0378 | 0.0174 | 0.0190 | 0.2419 | 0.1511 | 0.2283 | **0.8794** |

[1] The calculation of SSIM was described in Section 2.

## 4. Discussion

In this section, we will discuss the next three parts: the network structure, the experimental results and the loss function.

### 4.1. Analysis of the Network Structure

Three different network structures are trained to illustrate and analyze the advantages of our algorithm. Net1 is BSC-Net, Net2 keeps the network depth unchanged but increases the number of convolutional layers per stage and Net3 keeps the number of convolutional layers unchanged but increases the network depth. A single dataset is used as the training set, and the blended loss function proposed above is adopted. The experimental environment and experimental process are the same as above [42–47].

Figure 14 and Table 3 show how these three networks perform under same training settings and how long they take to process an image. From the table, it can be found that the depth of the network and quantity of the convolution layers both slow down the processing speed. With the image size increasing, this effect becomes more pronounced. Therefore, network structures that are excessively deep or complicated do not satisfy the real-time requirements. According to the quantitative analysis of the three network's results in Table 3, Net2 and Net3 can slightly increase the image's PSNR, but it is not apparent, and the SSIM value has not changed significantly. Combining the experimental data, we can conclude that the structure suggested in this paper takes into account both the performance and processing time.

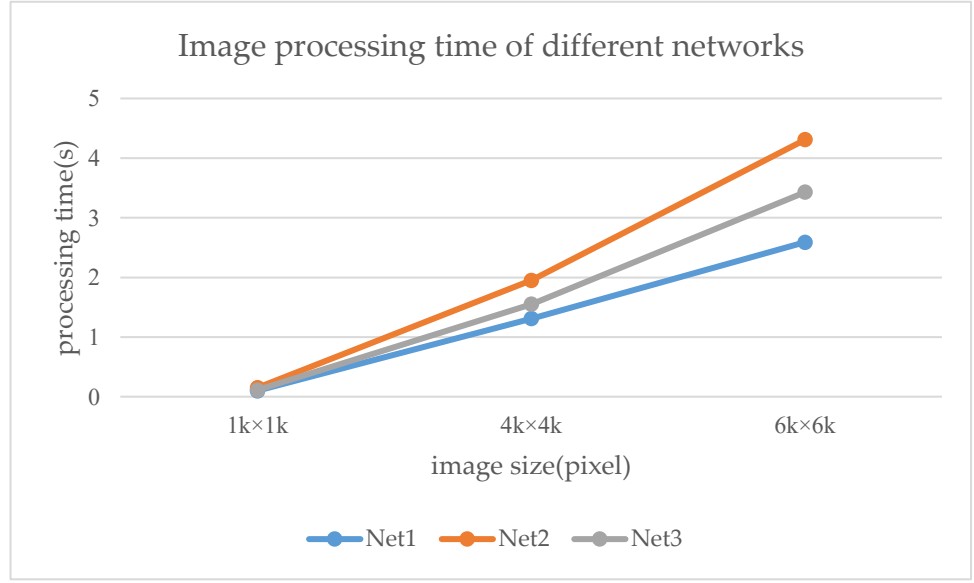

**Figure 14.** Image processing times of different networks.

**Table 3.** The quantitative results on different networks based on SSIM.

| Networks | SSIM | PSNR |
|---|---|---|
| Net1 | 0.933 | 64.90 |
| Net2 | 0.935 | 65.91 |
| Net3 | 0.935 | 67.50 |

### 4.2. Analysis of Results

In the comparative experiments, we analyzed the strengths and weaknesses of various methods and found that BSC-Net has the best processing effect. Unexpectedly, BSC-Net outperformed the morphological transform method that we used to form the training set. Furthermore, we discover that, in many complicated images, BSC-Net has a higher stability than morphological transform. In Figure 15, when the structural elements cannot be adapted to the images, the morphological transform causes error processing, such as suppressing the foreground or keeping the background. However, in the image processed by BSC-Net, there are no such problems.

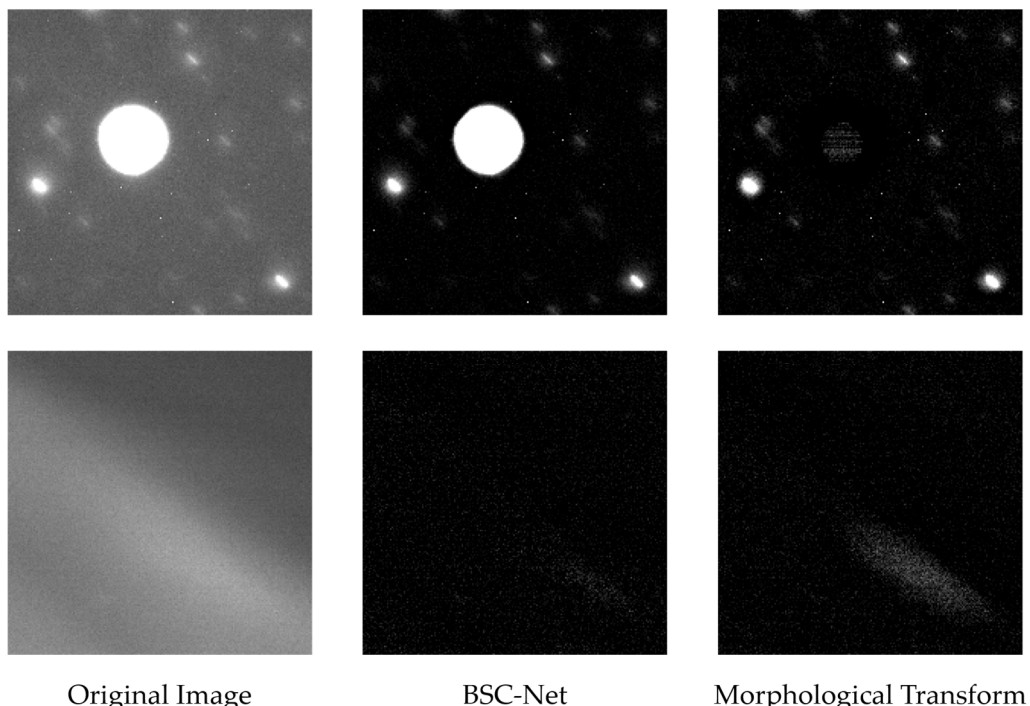

Original Image　　　　　BSC-Net　　　　Morphological Transform

**Figure 15.** Processing results of BSC-Net and the morphological transform.

This is because, despite the drawbacks of morphological transform, it can correctly process most of the images and provide high-quality training sets, which enable BSC-Net to learn a large amount of correct data. BSC-Net's numerous convolutional layers stand for various filters that can correspond to various targets, star points and stray light types. Compared with the limited number of structural elements in the morphological transform algorithm, the structure of BSC-Net will obviously have better stability. In the face of intricate real star images, BSC-Net will show stronger background suppression effects.

### 4.3. Analysis of Loss Function

In this paper, we propose the blended loss function of smooth_L1&SSIM, which plays an important role in the training of the network, mainly in accelerating the convergence of the loss function and preventing the gradient from exploding. We compare smooth_L1 with two other commonly used loss functions. As can be seen from Figure 16, under the same training conditions, the fluctuation of L2_LOSS is much larger than that of the other

two loss functions, the convergence speed is slow and gradient explosion is prone to occur. Smooth_L1 has the fastest decline, and at 200 epochs, the loss value drops to 0.38, while L1_LOSS is not as fast as smooth_L1 [48–52].

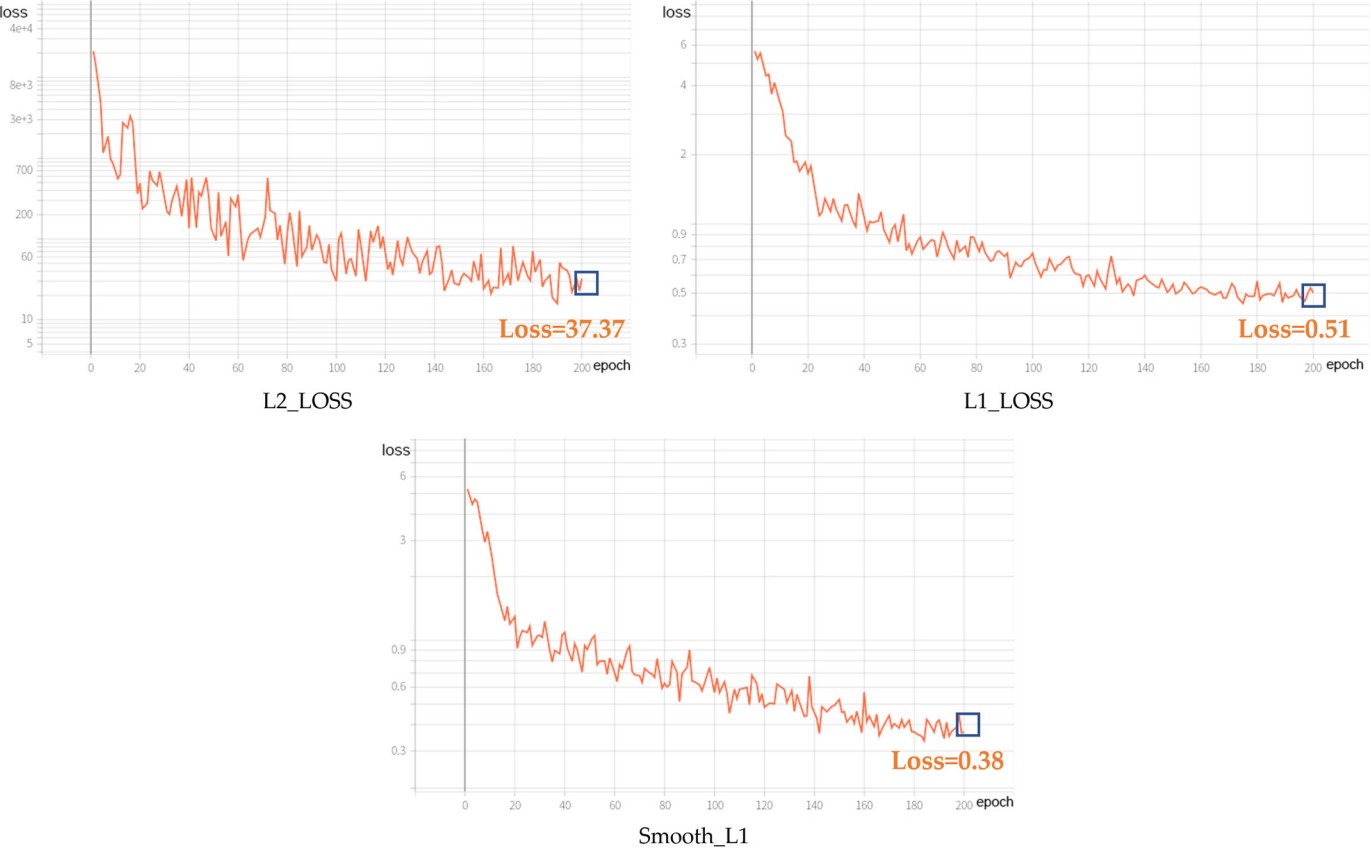

**Figure 16.** Variation curves of different loss functions. The blue boxes indicate the end point of the curves.

By training a single sample set with the proposed loss function and recording the SSIM value of the same image every 20 epochs, the results are shown in Figure 17. We discover that, as iterations are added, the SSIM value grows. Since the image quality is positively correlated with the SSIM value, the higher the SSIM value, the less distortion of the image. Therefore, we conclude that the blended loss function plays a positive role in network training and helps to improve the image's quality.

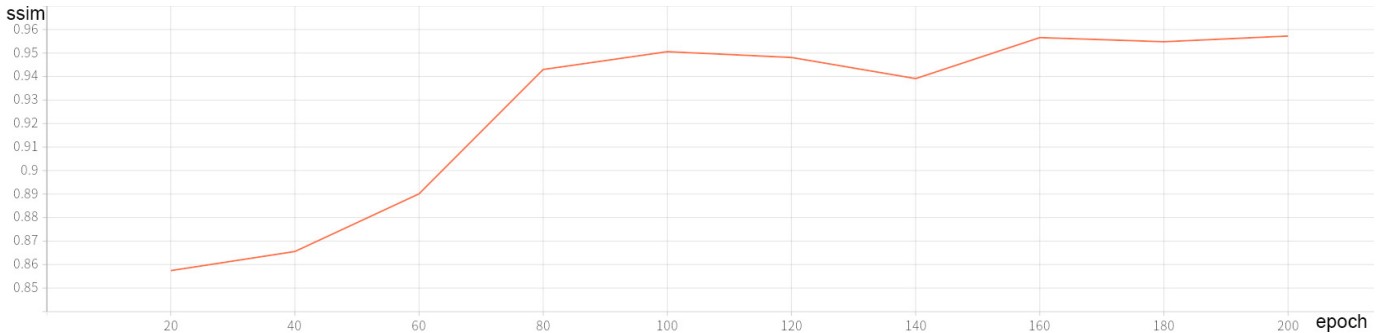

**Figure 17.** SSIM value variations curve during network training.

## 5. Conclusions

A background suppression algorithm for stray lights in star images is suggested in this paper. An end-to-end network structure, BSC-Net, is built to achieve background

suppression, and the blended loss function of smooth_L1&SSIM is utilized to avoid image distortion. The final network model is obtained by training on a wide number of real data. According to comparative experiments with traditional algorithms and deep learning algorithms, it is found that BSC-Net has a better robustness, and the SNR of the target has been enhanced the most. At the same time, on different test sets, the average PSNR improvement of BSC-Net reaches 42.61 dB, the average SSIM reaches 0.8713 and the image quality is excellent. Our research results not only prove the efficiency and practicality of BSC-Net but also provide a reference for the application of deep learning in this field. In the future, we plan to improve the algorithm and apply it to images collected by other devices.

**Author Contributions:** Conceptualization, Y.L. and Z.N.; methodology, Y.L.; software, Y.L. and H.L.; validation, Y.L.; formal analysis, Y.L.; resources, Q.S. and Z.N.; data curation, Q.S.; writing—original draft preparation, Y.L.; writing—review and editing, Z.N.; visualization, Y.L. and supervision, H.X. All authors have read and agreed to the published version of the manuscript.

**Funding:** This research was funded in part by the Youth Science Foundation of China (Grant No.61605243).

**Data Availability Statement:** Part of the data and code can be found at the following link: https://github.com/yblee999/Background-Suppression.git (accessed on 20 March 2022).

**Acknowledgments:** The authors would like to thank all of the reviewers for their valuable contributions to our work.

**Conflicts of Interest:** The authors declare no conflict of interest.

### Appendix A

Parameter settings of Dhyana 4040: The exposure time is about 100 ms. The PSF is two-dimensional Gaussian and in an area f radius of 1.5–2.0 pixels; the energy concentration degree is 80%. The field of view is $3 \times 3$ degrees, the image size is $4096 \times 4096$, and the pixel size is about 2.6 arcsec per pixel. The sky location captured by the device is random. The pixel value is 65,535.

Background RMS: The RMS calculation function in python port "sep" is used to obtain the RMS values of different images.

The references and available codes of each comparative experiment can be found in the following link: https://github.com/yblee999/Background-Suppression (accessed on 20 March 2022). The codes are for reference only, and the parameters used in the experiment will also be explained in the code.

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
