# Peer review of "BSC-Net: Background Suppression Algorithm for Stray Lights in Star Images"

_remotesensing, doi:10.3390/rs14194852_

Round 1
Reviewer 1 Report
The paper suggests a background suppression algorithm for stray lights in star images using an end-to-end network structure.
The paper is interesting and I enjoyed reading it.
In Equation no. 1, it is unclear what is the difference between B(i,j) and N(i,j). Please clarify.
In figure 4, please explain each of the four squares. What has been achieved in each down sampling?
In Equation no. 11, please explain the motivation to handle differently |x| values that are less than 1 and |x| values that are not less than 1.
I did not succeed to find in the paper whether the results are for Solar-type Stars, Hot Blue Stars, Red Dwarf Stars or other types of stars. Is there a distinction in your model for different stars?
When you cite a paper, do not write "the literature", but rather the name of the author/authors of the paper.
In [10]-[14], you survey several works of image segmentation. The survey is very interesting and essential. However, the paper cites only the newest paper and does not mention that automatic segmentation is more than 20 years old. I would encourage you to cite also Y. Wiseman & E. Fredj, "Contour Extraction of Compressed JPEG Images", ACM - Journal of Graphic Tools, Vol. 6(3), pp. 37-43, 2001, available online at: https://u.cs.biu.ac.il/~wiseman/jgt2001.pdf and also Cheng, H. D., Jiang, X. H., Sun, Y., & Wang, J. "Color image segmentation: advances and prospects", Pattern recognition, Vol. 34(12), pp. 2259-2281, 2001 which both suggest an approach for automatic segmentation.
The English of the paper must be improved. Especially, please pay attention to the article "a".
Reviewer 2 Report
Thank you for this contribution. This is an interesting and timely manuscript. The conducted analysis and proposed method for stray light suppression is typically standard and falls within the expected work from such a publication and hence the work merits publication. As such, the authors are invited to properly address the following items:
1. In general, the introduction is light and does not represent the state of the art in this domain. The amount of works in this area continues to rapidly rise. The authors are advised to strengthen their literature review section with supplementary material. Perhaps the addition of 1-2 pages can help strengthen this section.
2. For a ML-based work, a question arises as to how our readers can benefit from the developed models. Thus, the authors are advised to consider providing their code and database for interested researchers to extend and benefit from this work. For example, the authors may option to upload this database into Mendeley or attach it to this paper. The same can be done for the code.
3. The selection of performance metrics is not clear. For example, both Alavi et al (2021) [https://doi.org/10.1007/s44150-021-00015-8] and Botchkarev (2019) [https://doi.org/10.28945/4184] present a detailed review on the proper selection of metrics.
Reviewer 3 Report
The authors present a new, neural-network based method to suppress images background and enhance star-like objects' signal to noise ratio.
The results look very promising but, as for other similar papers, I notice the lack of comparison of the method effectiveness compared to that
of canonical astronomical images analysis tools. I guess the involvement in the work of an (optical) astronomer would improve their valuable work.
In fact the astronomical images processing approach to use depends on the aim one wants to achieve. A "nice" to look image could only
be relevant for presentation purposes, but "could" not add any scientifically relevant information.
Of course in some cases stray light could cause a severe image degradation and a methodology like that presented here could improve the scientific content of an image, but in other cases (e.g. when a precise estimate of the source magnitude/flux is not that important) a standard image cleaning (or none at all) could be sufficient.
There are several packages that can be tested and the authors are free to use any, but I would suggest Source Extractor (http://www.astromatic.net/software/sextractor) or its python port "sep" (https://github.com/kbarbary/sep). This code can also produce the objects' pixel mask (called "binarized image" by the authors). I consider this test useful and the results could complement those reported in the manuscript.
That said, I list here the major points that the authors should address/clarify before the paper can be published:
- It is not clear how the used reference images were produced. Do they refer to a single camera/instrument set-up?
The reader should be informed about their basic characteristics: pixel size, typical PSF, size / field of view (are they all the same?), limiting magnitude/flux/pixel value, background RMS, objects' density.
- As the method is particularly relevant for detecting weak objects (with respect to the background), it would be important to include
some specific tests for these cases (i.e. excluding high S/N objects from the "quantitative results").
- Image analysis issues related to "space debris" is apparently one of the main aspects the proposed method wants to address, but it is
only mentioned here and there. It seems mainly ignored in the test images as "elongated" objects (tracks) are not included (but eventually in Fig. 11).
I suggest specific tests for these cases and the results included in the manuscript. Alternatively this aspect can be investigated in more detail in a follow-up paper.
- The authors do not provide any reference to a tool (source code, pipeline, webtool or else) the reader can use to test the proposed method.
The diagram presented in Fig. 3 is fine, but (at least) some pseudo-code involved in the various steps could give a better understanding of the procedure.
This could go in an Appendix or, again, can be presented in a follow-up paper; but it should be clearly stated.
Detailed comments are given in the attached annotated PDF file.

Round 2
Reviewer 1 Report
The authors made a decent effort and the paper is certainly publishable so I would recommend accepting the paper.
Author Response
Thanks for your advice and work.
Reviewer 2 Report
.
Author Response
Thanks for your advice and work. We will check the English spelling and grammar further.
Reviewer 3 Report
The authors have implemented / replied most of the comments I reported in my first review. However I have some additional notes that should be taken into account before the manuscript gets published.
- My notes about the used images characteristics have been partially misinterpreted.
1. Of course the typical (simplified) PSF is a 2-d Gaussian, but its "size in terms of pixels" can vary (depending on various parameters). Testing the tool on images having various PSF sizes would help (and convince) the reader that the methodology is working well for the various situations.
2. Field of view (say 0.5x0.5 degrees) and pixel size (say 1 arcsec per pixel) defines how "wide and fine" a field in the sky is observed. Again investigating the various situations would help to characterize the effectiveness of the algorithm.
3. The objects density (how many "stars and other objects") are present in the image depends not only from the instrument sensitivity, but also from how long and "where in the sky" one is observing. Again, investigating the various cases is relevant.
So, it is completely understandable that the authors do not take all this into account for the present work, but I suggest that this is clearly stated and that further investigations on a this "variety" of sky images should/could be considered in a further work before claiming that the algorithm is working well for any type of image (which could well be).
- Because the BSC-Net architecture is clearly a U-net, I would recommend citing (and eventually highlighting differences) Ronneberger at al. 2015 (https://arxiv.org/abs/1505.04597)
- In the attached PDF some additional, minor comments are reported, but please note that these are "not" exhaustive and I would suggest a careful rereading, possibly by an English mother language person.

Author Response
Thank you so much for your detailed review. Please see the attachment.
